# Improving Rice Leaf Shape Using CRISPR/Cas9-Mediated Genome Editing of *SRL1* and Characterizing Its Regulatory Network Involved in Leaf Rolling through Transcriptome Analysis

**DOI:** 10.3390/ijms241311087

**Published:** 2023-07-04

**Authors:** Yue Han, Jinlian Yang, Hu Wu, Fang Liu, Baoxiang Qin, Rongbai Li

**Affiliations:** State Key Laboratory for Conservation and Utilization of Subtropical Agro-Bioresources, College of Agriculture, Guangxi University, Nanning 530004, China

**Keywords:** rice, CRISPR/Cas9, *SRL1*, transcriptome, leaf rolling

## Abstract

Leaf rolling is a crucial agronomic trait to consider in rice (*Oryza sativa* L.) breeding as it keeps the leaves upright, reducing interleaf shading and improving photosynthetic efficiency. The *SEMI-ROLLED LEAF 1* (*SRL1*) gene plays a key role in regulating leaf rolling, as it encodes a glycosylphosphatidylinositol-anchored protein located on the plasma membrane. In this study, we used CRISPR/Cas9 to target the second and third exons of the *SRL1* gene in the indica rice line GXU103, which resulted in the generation of 14 T_0_ transgenic plants with a double-target mutation rate of 21.4%. After screening 120 T_1_ generation plants, we identified 26 T-DNA-free homozygous double-target mutation plants. We designated the resulting *SRL1* homozygous double-target knockout as *srl1-103*. This line exhibited defects in leaf development, leaf rolling in the mature upright leaves, and a compact nature of the fully grown plants. Compared with the wild type (WT), the T_2_ generation of *srl1-103* varied in two key aspects: the width of flag leaf (12.6% reduction compared with WT) and the leaf rolling index (48.77% increase compared with WT). In order to gain a deeper understanding of the involvement of *SRL1* in the regulatory network associated with rice leaf development, we performed a transcriptome analysis for the T_2_ generation of *srl1-103*. A comparison of *srl1-103* with WT revealed 459 differentially expressed genes (DEGs), including 388 upregulated genes and 71 downregulated genes. In terms of the function of the DEGs, there seemed to be a significant enrichment of genes associated with cell wall synthesis (*LOC_Os08g01670*, *LOC_Os05g46510*, *LOC_Os04g51450*, *LOC_Os10g28080*, *LOC_Os04g39814*, *LOC_Os01g71474*, *LOC_Os01g71350*, and *LOC_Os11g47600*) and vacuole-related genes (*LOC_Os09g23300*), which may partially explain the increased leaf rolling in *srl1-103*. Furthermore, the significant downregulation of BAHD acyltransferase-like protein gene (*LOC_Os08g44840*) could be the main reason for the decreased leaf angle and the compact nature of the mutant plants. In summary, this study successfully elucidated the gene regulatory network in which *SRL1* participates, providing theoretical support for targeting this gene in rice breeding programs to promote variety improvement.

## 1. Introduction

Rice, as an essential food crop, serves as a fundamental source of sustenance for millions of people worldwide [1]. The yield of rice is influenced by various factors, with the photosynthetic efficiency of leaves being one of the most significant factors [2]. Moderate leaf rolling enables the leaves to remain upright, reducing interleaf shadow, improving photosynthetic efficiency, and increasing rice yield. Therefore, moderate leaf rolling is recognized as a crucial agronomic characteristic in rice breeding [3,4,5]. In recent years, several genes responsible for controlling leaf rolling have been identified, providing a promising foundation for future breeding efforts to induce rice leaf rolling [6].

Although leaf rolling is a straightforward and intuitive agronomic trait, its regulatory mechanisms are complex. Mutations in genes related to leaf development and the impacts of diverse environmental conditions can both result in leaf rolling. Leaf development is an intricate process involving the formation of adaxial–abaxial axis, bulliform cells, sclerenchymatous cells, cuticle, and epidermal cells [7]. Mutations in genes involved in these processes can lead to abnormal leaf development and the occurrence of the rolled-leaf phenotype. Currently, more than 30 genes affecting leaf rolling have been cloned, and the majority of these genes operate by regulating the number, size, or arrangement of bulliform cells. *SEMI-ROLLED LEAF 1* (*SRL1*) is a gene involved in the regulation of leaf rolling and encodes a putative glycosylphosphatidylinositol-anchored protein located on the plasma membrane. It is expressed in various tissues of rice and acts as a negative regulator of genes encoding vesicular H^+^-ATPase subunits and H^+^-pyrophosphatase, inhibiting the formation of bulliform cells and regulating leaf rolling [8]. *CURLED LEAF AND DWARF 1* (*CLD1*) is allelic to *SRL1*, which is closely associated with cell wall formation, epidermal integrity, and the maintenance of water homeostasis. *cld1* mutants exhibited a considerable decrease in the content of cellulose and lignin within the secondary cell wall of the leaf when compared with the wild type (WT). The loss of *CLD1* function could lead to defective leaf epidermal development, reduced water retention capacity in leaves, and eventually rolled leaves [9]. *RICE OUTER CELL SPECIFIC 8* (*ROC8*), a homologous structural domain leucine zipper class IV gene, is involved in the negative regulation of bulliform cell size. The overexpression of *ROC8* reduced the number and size of bulliform cells, resulting in leaf adaxial rolling. Conversely, the knockdown of *ROC8* increased the number and size of bulliform cells, leading to leaf abaxial rolling [10]. *ROC5* and *ROC8* exhibit similar biological functions in regulating leaf rolling and both have additive effects in function, and can form ROC8–ROC5 heterodimers to jointly regulate leaf rolling [11]. Additionally, *ROLLED AND ERECT LEAF 1* (*REL1*) [12], *ABAXIALLY CURLED LEAF 2* (*ACL2*) [13], *ZINC FINGER HOMEODOMAIN 1* (*OsZHD1*) [14], and *REL2* [15] have also been found to affect leaf rolling by regulating the number, size, or arrangement of bulliform cells.

In addition to leaf rolling caused by changes in bulliform cells, variations in cellulose and lignin content can also contribute to leaf rolling. *NARROW AND ROLLED LEAF 2* (*NRL2*) [16], *OsMYB103L* [17], and *ROLLING-LEAF 14* (*RL14*) [18] are involved in regulating the synthesis of cellulose and lignin, thereby affecting the morphology and mechanical strength of leaves. Moreover, changes in cellulose and lignin content also play a crucial role in the synthesis of secondary cell walls, which is essential for water transport in leaves. Alterations in cellulose and lignin synthesis can also lead to abnormal bulliform cell morphology that ultimately causes leaf rolling [18]. In addition, *SRL10* [19], *SHALLOT-LIKE 1* (*SLL1*) [20], *CONSTITUTIVELY WILTED 1* (*COW1*) [21], *ADAXIALIZED LEAF 1* (*ADL1*) [22], *CURLY FLAG LEAF 1* (*CFL1*) [23], *NRL1* [24] and *ROLLED LEAF 17* (*RL17*) [25] also regulate the leaf rolling process through several different regulatory networks. In conclusion, the regulatory mechanism of leaf rolling is complex and influenced by several factors.

The CRISPR/Cas system is widely recognized as a mainstream gene editing tool for plant improvement. The guide RNAs effectively guide the Cas9 enzyme to create accurate cuts in the DNA double helix, which results in double-stranded breaks. These breaks can cause targeted gene mutations, such as gene knockout, when the DNA is repaired. This process ultimately achieves the desired modification of the genomic DNA [26]. With the continuous improvement of CRISPR/Cas9 editing technology, it is now possible to simultaneously target multiple sites, providing new methods for studying traits controlled by multiple genes and gene families [27]. The precise genome editing enabled by CRISPR/Cas9 has the potential to greatly enhance the quality and yield of agricultural production, while also facilitating the creation of new and improved varieties of rice with enhanced traits [28,29,30,31,32]. In recent years, the beneficial integration of CRISPR/Cas9 and high-throughput transcriptome sequencing technology (RNA-seq) has opened up new possibilities for investigating gene function [33,34]. Simultaneously, with the development of molecular biology technology, RNA-seq has been widely used in the study of gene expression levels and regulatory networks [35,36,37]. Combined with typical bioinformatics analysis methods such as differential expression analysis, gene network screening, and function enrichment, transcriptome data can comprehensively and systematically identify gene expression regulatory networks at the RNA level, providing a new direction for the further study of target genes [38].

Although numerous genes have been linked to leaf rolling, there are still gaps in our comprehension of the molecular mechanisms that regulate leaf morphology. The previous study employed microarray analysis to investigate the network controlled by *SRL1*. However, it only focused on the differential gene expression in the cells abutting the midrib at the adaxial surface of leaf blades, which develop into epidermal cells in wild-type plants and probably transform into bulliform cells in *srl1-1*. Due to the differences in sampling locations and limitations of microarray technology, this could potentially result in an incomplete understanding of the regulatory mechanisms of *SRL1* in leaf development. In order to address the limitations of previous research and obtain a more comprehensive understanding of the regulatory network in leaf development involving *SRL1*, we used CRISPR/Cas9 to create a double-target knockout mutant of *SRL1* in the indica rice background and compared the gene expression profiles between the mutant and control. The study primarily focused on the key genes influenced by *SRL1* in vacuole development and cell wall synthesis, as well as the potential pathways through which it may affect changes in plant morphology.

## 2. Results

### 2.1. Construction of CRISPR/Cas9 Knockout Vector

The *SRL1* gene is composed of eight exons and seven introns, with its full-length complementary DNA (cDNA) sequence spanning 1871 bp and encoding a protein consisting of 441 amino acids. To completely disrupt the function of *SRL1* and enhance the mutation efficiency, two knockout target sites were selected within the gene. Specifically, these target sites were located in the second exon (Target 1: 5′-GCAGGTGTGCCCAATTCATT-3′) and the third exon (Target 2: 5′-GGCGGTTTGACAAGCTACGC-3′) (Figure 1A). Then, the U6a-sgRNA and U6b-sgRNA expression cassettes were obtained using overlapping PCR amplification (Figure 1B), and the pY-CRISPR/Cas9 Pubi-H knockout vector was successfully constructed (Figure 1C).

### 2.2. Selection of T-DNA-Free Double-Target Homozygous Mutant Rice Lines

Calluses of the indica rice line GXU103 were infested with *Agrobacterium tumefaciens* carrying a double-target CRISPR/Cas9 vector. Positive mutant plants were selected using specific primers, hyg-F/hyg-R (Appendix A), which contained a hygromycin resistance marker, and a total of 14 positive plants were obtained (Figure 2B). For each T_0_ individual plant, the targets were amplified by PCR and identified through sequencing. At the first target position, there were five homozygotes, seven heterozygotes, and two WTs. Similarly, at the second target position, there were four homozygotes, five heterozygotes, and five WTs (Table 1). Three plants were identified as double-target mutants (*GXU103-X* represents different mutant types), resulting in a double-target mutation rate of 21.4%. *GXU103-1* had a single-base insertion at the first target site and a 3 bp deletion at the second target site. *GXU103-5* had a 4 bp deletion at the first target site and a single base insertion at the second target site, while *GXU103-12* had a 5 bp deletion at the first target site and a single-base insertion at the second target site (Figure 2A). The plants exhibiting these three distinct types of double-target mutations, which were discovered in T_0_ generation, were used in subsequent experiments.

Self-pollinated seeds from all mutant lines in the T_0_ generation were harvested and referred to as T_1_ generation seeds. Using primers Cas9-F/Cas9-R (Appendix A), 120 T_1_ generation plants were detected, out of which 73 plants were identified as T-DNA-free. Among these 73 plants, a total of 26 were discovered to be homozygous for double-target mutations, and they were subsequently named *srl1-103* (Figure 2C). The sequencing results show that the double-target mutation types in the T_1_ generation of *srl1-103* were all derived from *GXU103-X* in the T_0_ generation, confirming that the double-target mutation types detected in the T_0_ generation were stably inherited in the T_1_ generation of *srl1-103*. Three distinct double-target mutation types from the T_2_ generation of *srl1-103* were selected, and 100 individual plants were cultivated for each genotype. These 300 individual plants all exhibited defects in leaf development, with upright rolled leaves and a more compact plant type compared with WT. Finally, *GXU103-1* mutant plants from the T_2_ generation of *srl1-103* were selected for subsequent experimental studies.

### 2.3. Analysis of Leaf Physiological Morphology and Agronomic Traits

In order to further explore the variations in traits between the mutant line *srl1-103* and the wild-type GXU103, the T_2_ generation of *srl1-103* was utilized as the experimental materials for analysis. When investigating GXU103 and *srl1-103* at maturity, it was observed that the leaves of GXU103 were naturally pendulous, while *srl1-103* displayed an overall erect and clustered growth status with inwardly rolled leaves (Figure 3A,B). Through the leaf cross-section, it was evident that the leaves of GXU103 are relatively flat, with slightly upward leaf blade edges. In contrast, the leaf blades of *srl1-103* exhibited significant inward rolling (Figure 3C). Further experimental observations revealed that the bulliform cells on the leaves of GXU103 were neatly arranged, whereas those of *srl1-103* were larger and appeared irregularly arranged between two adjacent small vascular bundles. The total number of epidermal cells and bulliform cells was similar between GXU103 and *srl1-103* (Figure 3D).

To examine the impact of *SRL1* gene knockout on rice agronomic traits, an investigation was conducted on mature GXU103 and *srl1-103* plants. The results show that there were no significant differences in plant height, length of flag leaf, number of panicles, length of panicle, seed setting rate, and 1000-grain weight (Figure 3E,F,I–L). However, compared with GXU103, the flag leaf width of *srl1-103* was 12.6% narrower, showing a significant difference (Figure 3G). Surprisingly, the leaf rolling index (LRI) of *srl1-103* reached 57.12%, which was 48.77% higher than GXU103, indicating a highly significant difference (Figure 3H). The above results indicate that the knockout of *SRL1* altered the morphology of GXU103 leaves, resulting in increased leaf rolling and upright growth.

### 2.4. Transcriptome Analysis of the Wild-Type GXU103 and the Mutant Line srl1-103

To uncover the regulatory network involved in leaf rolling associated with *SRL1*, RNA-seq analysis was performed on mature leaves of the wild-type GXU103 and the T_2_ generation of *srl1-103*. For each group, the samples were collected from three independent plants. After filtering out the ineligible reads from the raw data, six libraries were obtained. Each library consisted of over 18,852,288 reads, with more than 15,441,394 reads aligning to the reference genome at a unique comparison rate higher than 81.80% (Table 2). The results indicate that high-quality sequencing data were obtained from RNA-seq. Additionally, the correlation between the samples was evaluated by the Pearson’s correlation coefficient (PCC), revealing that the wild-type GXU103 exhibited a PCC value above 0.94, while the mutant line *srl1-103* displayed a PCC value higher than 0.96. In general, all the results indicate that the high-quality sequencing data obtained from RNA-seq had a high level of confidence and could be used for subsequent transcriptome analysis (Figure 4A).

In *srl1-103*, a total of 459 differentially expressed genes (DEGs) were detected by RNA-seq, of which 388 genes were significantly upregulated, while 71 genes were downregulated (Appendix A). Among them, the expression of *SRL1* (*LOC_Os07g01240*) was notably decreased in *srl1-103*, suggesting the successful knockout of *SRL1* through CRISPR/Cas9 (Figure 4B). To investigate the involvement of DEGs in the development of the rolled leaf phenotype, a gene ontology (GO) analysis was performed on these genes. The analysis unveiled that DEGs were significantly enriched in various biological pathways, including cell structure, mitosis, phytohormones, epidermal development, stress response, cellular homeostasis, and cellular metabolism (Figure 4C).

### 2.5. Impact of SRL1 Knockout on Cellular Structure

To further understand the mechanism of *SRL1* regulation in leaf rolling, we focused on the genes related to the cell wall synthesis pathway (Table 3). Plant cell walls are mainly composed of cellulose, hemicellulose, lignin, and pectin. They provide mechanical support for plant tissues and play an important role in the development of plant morphology. In the transcriptome data, there were aberrant expressions of certain genes involved in cell wall synthesis. For example, *LOC_Os08g01670* (*RICE PECTIN METHYLESTERASE INHIBITOR 28*, *OsPMEI28*), which exhibits pectin methyl esterase inhibitor activity and plays a key role in regulating the level of pectin methylation, was significantly upregulated in *srl1-103*. *LOC_Os05g46510*, which encodes a putative polygalacturonase (PG) with pectin-degrading effects, was also similarly significantly upregulated. Glycosyl hydrolase (GH) is one of the most widely distributed protein hydrolases in plants, participating in multiple physiological processes such as plant cell wall remodeling, lignification, and phytohormone activation. It was found that *LOC_Os04g51450*, *LOC_Os10g28080*, *LOC_Os04g39814*, *LOC_Os01g71474*, *LOC_Os01g71350*, and *LOC_Os11g47600*, each of which encodes a putative glycosyl hydrolase, were likely to be involved in plant cell wall remodeling and were significantly upregulated in *srl1-103*. In addition to genes related to the cell wall synthesis pathway, significant changes were found in vacuole-related genes. Previous studies on *SRL1* have suggested that it negatively regulates the expression of vacuole-related genes, which affect the formation of bulliform cells. The transcriptome data reveal that *LOC_Os09g23300* (*VACUOLAR IRON TRANSPORTER 2*, *OsVIT2*), which encodes a vacuolar membrane transporter protein responsible for ion transport processes, was significantly upregulated in *srl1-103*. The differential expression of genes related to cell wall synthesis and vacuolar function might be involved in the regulation of leaf rolling by *SRL1*.

### 2.6. Impact of SRL1 Knockout on Plant Type and Leaf Angle

In comparison to WT, *srl1-103* exhibited a more compact plant type characterized by smaller leaf angles and upright rolled leaves. To investigate the regulatory mechanism of plant type change in *srl1-103*, further analysis was conducted on differential genes related to the regulation of plant type (Table 3). It was shown that *LOC_Os08g44840* (*SLENDER GRAIN*, *SLG*), which encodes a BAHD acyltransferase-like protein involved in the regulation of leaf angle, was significantly downregulated in *srl1-103*. A previous study showed that the downregulation of *LOC_Os08g44840* led to a decrease in leaf angle and a more upright leaf in rice. Therefore, it could be inferred that the significant downregulation of *LOC_Os08g44840* in *srl1-103* was likely the main reason for the reduced leaf angle and compact plant architecture.

## 3. Discussion

Bulliform cells are a special type of epidermal cells that exist on the leaves of numerous plants. Typically, these cells are located in the upper epidermis or near the adaxial surface of the leaf, and display a large, empty, colorless, and bubble-shaped form, appearing between the two veins. Changes in the number, size, or arrangement of bulliform cells can result in leaf rolling. A previous report indicated that the leaf rolling phenotype of *srl1-1* was caused by the enhanced formation of bulliform cells at the adaxial cell layers [9]. According to our data, the knockout mutant *srl1-103* displayed larger and irregularly arranged bulliform cells in its leaves compared with WT (Figure 3D), which may be one of the reasons for the rolling of leaves. This finding aligns with previous reports highlighting the significant role of *SRL1* in the formation of bulliform cells. Statistical analysis was performed on several agronomic traits of *srl1-103* (Figure 3E–L). The results reveal that the knockout of *SRL1* significantly influenced two agronomic traits, the width of the flag leaf and LRI. In the *srl1-103* mutant, there was a noticeable reduction in the width of the flag leaf compared with WT, which could potentially affect its photosynthetic capacity and overall growth. Additionally, a significant alteration in LRI was observed in *srl1-103*, which reflects the degree of leaf rolling. The change in LRI suggests that leaf development has been disrupted, potentially affecting the plant’s ability to efficiently absorb sunlight and exchange gases with the environment. Interestingly, while there were significant changes in the flag leaf width and LRI, other agronomic traits such as plant height, number of panicles, and 1000-grain weight remained unaffected by the knockout of *SRL1*. This suggests that *SRL1* plays a specific role in certain aspects of leaf development and morphology, while not exerting an influence on the overall plant structure or reproductive characteristics.

As an important agronomic trait that is easily observable during the growth of rice, leaf rolling has garnered considerable attention from breeders [5]. The mechanisms of leaf rolling are complex, typically involving the development of bulliform cells and cell wall formation [39]. In this study, it was found that genes related to cell wall synthesis and vacuolar functions were abnormally expressed in *srl1-103*, which may be the primary cause of the altered leaf morphology. To better understand the potential pathways associated with the *SRL1* regulation of leaf rolling and plant type, the results of transcriptome data predictions were visualized (Figure 5). Among the DEGs, *LOC_Os08g01670* (*OsPMEI28*) exhibits pectin methylesterase inhibitor activity, and the overexpression of *OsPMEI28* has been shown to decrease cell wall thickness and the content of cell wall components [40]. *LOC_Os05g46510* encodes a putative polygalacturonase enzyme, which participates in cell wall biosynthesis by degrading the pectin within cellular components, consequently affecting the mechanical strength of the cell wall. *LOC_Os04g51450*, *LOC_Os10g28080*, *LOC_Os04g39814*, *LOC_Os01g71474*, *LOC_Os01g71350*, and *LOC_Os11g47600* encode presumed glycosyl hydrolases with the ability to hydrolyze polysaccharides within cell walls. In general, the three types of genes related to cell wall synthesis were significantly upregulated in *srl1-103*, indicating that *SRL1* may participate in cell wall formation by regulating the expression of these genes, thereby influencing leaf morphology. This finding is similar to previous studies on the regulatory mechanism of *CLD1/SRL1* [9].

A previous study on *SRL1* indicated that some vacuole-related genes are significantly upregulated in *srl1-1*, suggesting that *SRL1* negatively regulates the expression of vacuole H^+^-ATPase subunit and H^+^-pyrophosphatase genes to inhibit the formation of bulliform cells [8]. However, only one vacuole-related gene, *LOC_Os09g23300* (*OsVIT2*) [41], was found to be significantly upregulated in *srl1-103* in the transcriptome data. Leaf angle, which represents the inclination between the leaf blade and the vertical culm, plays a pivotal role in determining the overall plant architecture. Numerous studies have shown that the degree of the leaf angle is mainly associated with the biosynthesis and signal transduction pathways of brassinosteroid (BR), indole-3-acetic acid (IAA), and gibberellin (GA). Moreover, the interaction among these plant hormones also regulates the leaf angle to some extent [42,43,44]. The signaling pathway of BR is crucial for regulating the leaf angle in rice and involves the action of multiple genes. BRI1, encoded by the BR receptor kinase gene *OsBRI1* in rice, shares high sequence similarity with Arabidopsis BRI1 and plays various roles in the growth and development of rice. Deficits in the function of *OsBRI1* led to impaired BR signaling, resulting in reduced leaf angles and erect leaf traits in rice [45]. PRA2, encoded by the small G protein gene, can interact with BRI1 and inhibit its kinase activity, thereby negatively regulating BR signaling. The overexpression of *OsPRA2* resulted in decreased leaf angle and reduced sensitivity to exogenous BR. In contrast, suppressing the expression of *OsPRA2* through RNA interference led to an increase in leaf angle and enhanced sensitivity to exogenously applied BR [46]. In addition to the genes related to the BR signaling pathway, genes associated with BR homeostasis, such as *RAV6* and *SLG*, also affect the magnitude of the leaf angle. *RAV6* encodes a protein with a B3 DNA-binding domain, which affects rice leaf angle by mediating BR homeostasis, and the semidominant mutant *Epi-rav6* showed an increase in leaf angle [47]. *SLG* encodes a BAHD acyltransferase-like protein that serves as a crucial regulatory factor in maintaining BR homeostasis. It functions as homomers and regulates grain size and leaf angle in rice [48]. The transcriptome data indicate that *LOC_Os08g44840* (*SLG*) was significantly downregulated in *srl1-103*, possibly exerting a significant influence on plant architecture. This finding has not been previously emphasized in studies on *SRL1*. There may be potential interdependent regulatory mechanisms between *SRL1* and *SLG*, which necessitate further investigation to fully understand their comprehensive effects on the structure of rice.

Currently, there are few reports on the involvement of leaf rolling genes in stress tolerance pathways. These stress tolerance phenotypes are typically regulated by multiple genes, making their study more difficult. In this study, we found that the knockout of *SRL1* not only impacted leaf rolling and plant type, but it also affected the expression of some resistance genes (Appendix A). This suggests that the *SRL1* gene may be involved in certain stress tolerance regulatory networks. Chitinase is a disease course-related protein that holds significance in plant defenses against pathogens and insect pests. Moreover, it actively participates in the plant’s response to abiotic stress [49,50,51]. The transcriptome data show that *LOC_Os04g41680*, which encodes a protein precursor of the chit3-chitinase family, was significantly upregulated in *srl1-103* and may enhance disease resistance. In addition, *LOC_Os02g14440* and *LOC_Os07g48010*, presumed precursors of peroxidase, were significantly upregulated in *srl1-103*, which may accelerate the peroxidase clearance process and improve the tolerance to oxidative stress. Furthermore, the knockout of *SRL1* also affected the differential expression of some MYB transcription factors. MYB transcription factors, as one of the most abundant transcription factor families in plants, are widely involved in plant stress responses [52,53,54]. The overexpression of *LOC_Os04g43680* (*Osmyb4*) can enhance cold tolerance of *Arabidopsis thaliana* and affect the expression of multiple genes in cold stress response pathways [55]. Interestingly, the transcriptome data reveal a significant upregulation of *Osmyb4* in *srl1-103*, suggesting that it might enhance the cold tolerance of *srl1-103*. Similarly, *LOC_Os11g45740* (*JASMONIC ACID-INDUCIBLE RICE MYB GENE*, *JAmyb*) encodes an R2R3-type MYB transcription factor. The overexpression of this gene has been shown to increase salt tolerance in transgenic lines [56], and its expression was significantly upregulated in *srl1-103*. Several genes encoding disease course-related proteins, whose expression is induced by jasmonic acid, play important roles in biological and abiotic stresses, and were also differentially expressed. *LOC_Os01g28450* (*PATHOGENESIS-RELATED CLASS 1B*, *OsPR1b*) [57], *LOC_Os03g18850* (*JASMONATE INDUCIBLE PR10*, *JIOsPR10*) [58], and *LOC_Os12g36830* (*ROOT SPECIFIC RICE PR10*, *RSOsPR10*) [59] were found to be significantly upregulated in *srl1-103*, which may increase its corresponding resistance capabilities. Based on the transcriptome data, it is speculated that *SRL1* is potentially involved in the regulation of the expression of these resistance genes, which may be a new direction in the study of leaf shape and stress tolerance. Additionally, the potential off-target effects of CRISPR/Cas9 gene editing technology may affect the analysis results. In this study, the predicted maximum off-target score for target 1 in indica rice was 0.23, while for target 2, it was 0.144. This indicates that further experiments are still needed to validate the results of the transcriptome analysis.

In this study, we conducted transcriptomic analysis to investigate the regulatory mechanism of *SRL1* in rice, aiming to provide valuable insights for the further exploration of *SRL1*-related functions. The results indicate that *SRL1* may affect leaf development through the regulation of genes associated with cell wall synthesis, vacuole development, and leaf angle. This fills a significant gap in previous research and enhances our understanding of the regulatory network in which *SRL1* may be involved, making it clear and comprehensive. Furthermore, we also observed that the knockout of *SRL1* led to a significant upregulation of certain resistance genes, suggesting that *SRL1* may be involved in the regulatory network of stress tolerance. Under stressful conditions such as high temperature, high salinity, and drought, leaves tend to curl in order to reduce transpiration and minimize water loss, thus enhancing their tolerance. Whether leaf rolling under normal conditions contributes to resistance remains to be further investigated through methods such as resistance identification, gene expression analysis, and interaction analysis. With the continuous improvement of CRISPR/Cas9 editing technology, it is now possible to simultaneously target multiple sites, providing new methods for studying traits controlled by multiple genes and gene families in plants. In addition, CRISPR/Cas9 has demonstrated tremendous potential in the field of breeding. By modifying multiple genes in crops, it enables the simultaneous improvement of various traits, including resistance, quality, and yield, thus advancing the breeding process. The integration of traditional breeding techniques with gene editing technologies is emerging as a significant trend in breeding research, offering numerous opportunities and challenges for future agricultural production.

## 4. Materials and Methods

### 4.1. Plant Materials and Growing Environment

GXU103, an indica rice line developed by the Rice Research Institute of Guangxi University, possesses outstanding agronomic traits and has been utilized for conducting genetic transformation and gene editing experiments. All rice lines were cultivated in a mesh room with isolation conditions in Nanning, China, during the normal rice-growing seasons, and treated in accordance with the conventional planting management method. The materials used for phenotypic observation, agronomic trait statistics, and transcriptome analysis were from the T_2_ generation of *srl1-103*, derived from the *GXU103-1* mutant genotype. GXU103 was utilized as the wild-type control.

### 4.2. Construction of CRISPR/Cas9 Knockout Vectors

The partial *SRL1* sequence of wild-type GXU103 was amplified using specific primers (SRL-F/SRL-R), and two target sites were selected using the CRISPR-GE website (http://skl.scau.edu.cn/ (accessed on 16 September 2020)). Then, the two targets were introduced into the SgRNA expression cassettes driven by U6a and U6b promoters, using overlapping PCR methods with specific primers (gRT-SRL1/OsU6a-SRL1; gRT-SRL2/OsU6b-SRL1; U-F/gR-R; Pps-GGL/Pgs-GG2; Pps-GG2/Pgs-GGR). The knockout vector was constructed by mixing the purified expression cassette mixture with the pYCRISPR/Cas9PubI-H plasmid, followed by transformation into the healing tissue of GXU103 using the *Agrobacterium* strain EHA105.

### 4.3. Genotype Determination of T_0_ Generation and Screening for T-DNA-Free Plants

Genomic DNA was extracted from fresh leaves of each plant. T_0_-positive plants were detected using specific primers (hyg-F/hyg-R) with a hygromycin resistance marker. The target sequences were amplified using the specific primers SRL-F/SRL-R and identified by Sanger sequencing. T-DNA-free plants were screened by amplifying T-DNA isolates of the T_1_ generation using Cas9-F/Cas9-R primers.

### 4.4. Statistical Analysis of Agronomic Traits

The agronomic traits of wild-type GXU103 and the T_2_ generation of *srl1-103* were investigated through field surveys. Five individual plants from each line were used for agronomic trait measurements. The main traits evaluated included plant height, length of flag leaf, number of panicles, length of panicle, seed setting rate, 1000-grain weight, width of flag leaf, and leaf rolling index. Statistical analysis was performed using the software GraphPad Prism 8. Significant differences were determined by the Student’s *t* test.

### 4.5. Histology and Microscopy Observations

Flag leaves were fixed in 25% glutaraldehyde solution, dehydrated by graded ethanol solutions, and then embedded in low-temperature paraffin. Cross-sections (12 mm) were cut using a sectioning machine (Leice RM2235), spread and dried, dewaxed, rehydrated in graded ethanol solutions, stained with filtered 1% toluidine blue, and sealed with gum Arabic. Sections were microscopically examined and photographed (Zeiss Scope).

### 4.6. Leaf Rolling Index Analysis

Leaf rolling index (LRI) was determined for WT and mutant plants at the maturity stage using the natural state leaf edge distance (Ln) and the expanded state leaf edge distance (Lw). The LRI was calculated as: LRI (%) = (Lw − Ln)/Lw × 100.

### 4.7. Transcriptome Sequencing and Analysis

Transcriptional profiling was conducted using mature leaf tissues from wild-type and the T_2_ generation of *srl1-103*. The samples for the RNA-seq experiment were obtained from three independent WT and mutant plants. The total RNA was extracted using an RNeasy Plant Mini Kit (No.74904, QIAGEN, Hilden, Germany). RNA sequencing libraries were constructed using the NEB Next Ultra TM RNA Library Preparation Kit Illumina (NEB, Omaha, NE, USA) and subsequently sequenced on a NovaSeq 6000 to generate double-ended 150 bp reads. The raw data were filtered and quality-checked using the default parameters of FASTP (an ultra-fast all-in-one FASTQ preprocessor). Then, the clean data were compared to the rice reference genome (ref, Minghui genome) using the default parameters of Hisat2 (a fast spliced aligner with low memory requirements). Stringtie (enables improved reconstruction of a transcriptome from RNA-seq reads) was used to calculate gene expression levels, and DESeq2 (moderates estimation of fold change and dispersion for RNA-seq data) was utilized to identify differentially expressed genes between samples, with the criteria of fold change > 2 and FDR (false discovery rate) < 0.05. Subsequently, GO functional enrichment analysis for GO terms with a *p* value < 0.05 was performed using GOATOOLs (a Python library for gene ontology analyses).

## Figures and Tables

**Figure 1 ijms-24-11087-f001:**
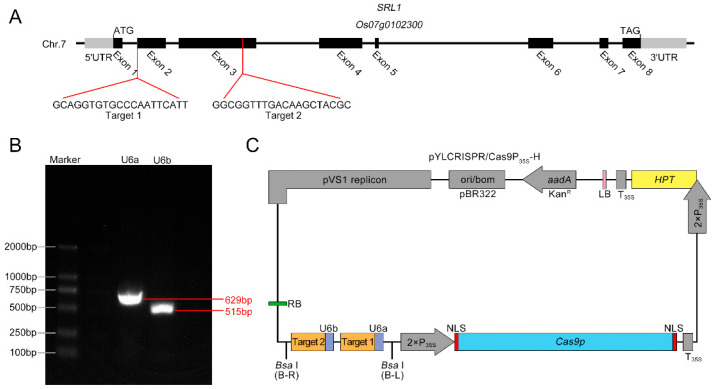
sgRNA design for *SEMI-ROLLED LEAF 1* (*SRL1*) target sequences and CRISPR/Cas9 vector construction. (**A**) The gene structure of *SRL1*, as well as the sequence and location of the targets. (**B**) The sgRNA expression cassettes for the two targets, U6a (629 bp) and U6b (515 bp). (**C**) Construction of CRISPR/Cas9 knockout vector.

**Figure 2 ijms-24-11087-f002:**
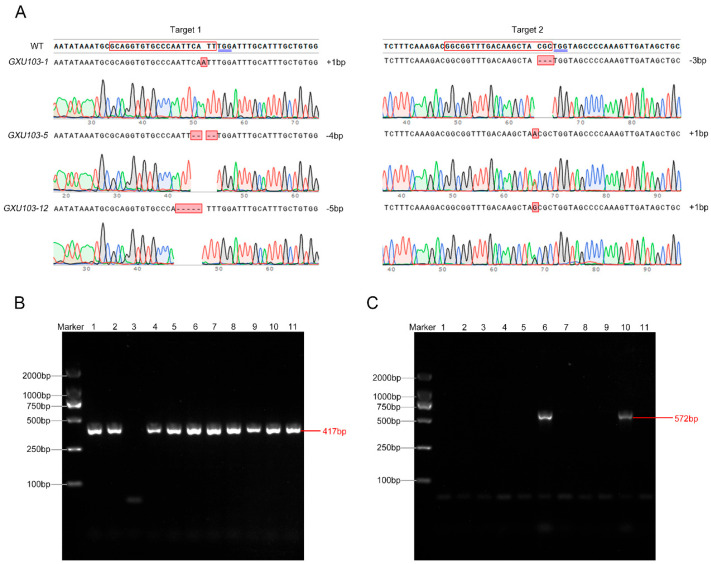
Target mutation types of T_0_ double-target mutant plants, and identification and screening of T_0_ and T_1_ generations. (**A**) Sequence schematic of the targets in *SRL1* and the types of double-target mutations. Targets are in red boxes, PAMs are underlined in blue, insertion and deletion nucleotides are in red boxes. (**B**) Amplification results of partial T_0_ generation plants using specific primers hyg-F/hyg-R (417 bp PCR product). (**C**) Exogenous screening of partial T_1_ generation using specific primers Cas9-F/Cas9-R (572 bp PCR product). Marker: 2000 bp. The numbers 1–11 represent the lanes. Wild type: WT.

**Figure 3 ijms-24-11087-f003:**
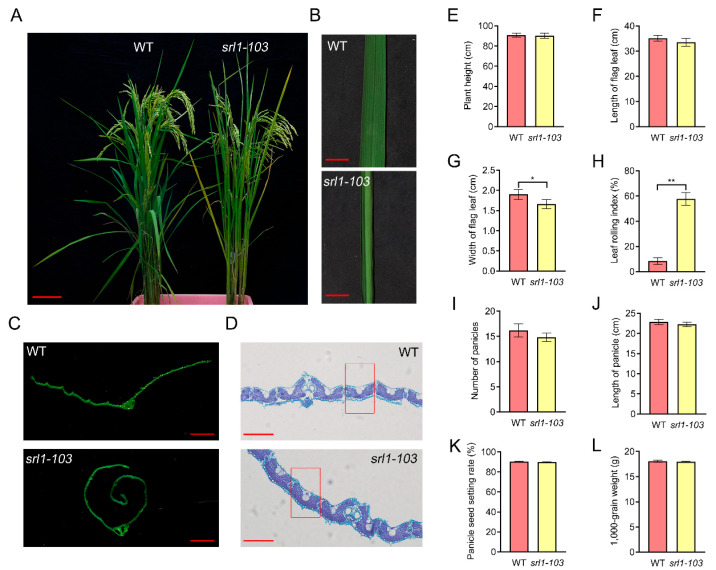
Physiological morphology and agronomic traits of the wild-type GXU103 and the mutant line *srl1-103*. (**A**) Plant at maturity, bar = 10 cm. (**B**) Frontal photograph of mature flag leaf, bar = 2 cm. (**C**) Section of mature flag leaf, bar = 2 mm. (**D**) Cross-section of leaf blade, bar = 100 μm. Bulliform cells between vascular bundles are in red boxes. (**E**–**L**) Agronomic traits of the wild-type GXU103 and the mutant line *srl1-103*: plant height, length of flag leaf, width of flag leaf, leaf-rolling index, number of panicles, length of panicle, seed setting rate, 1000-grain weight. Statistical analysis used Student’s *t* test to determine significant differences (* *p* < 0.05, ** *p* < 0.01).

**Figure 4 ijms-24-11087-f004:**
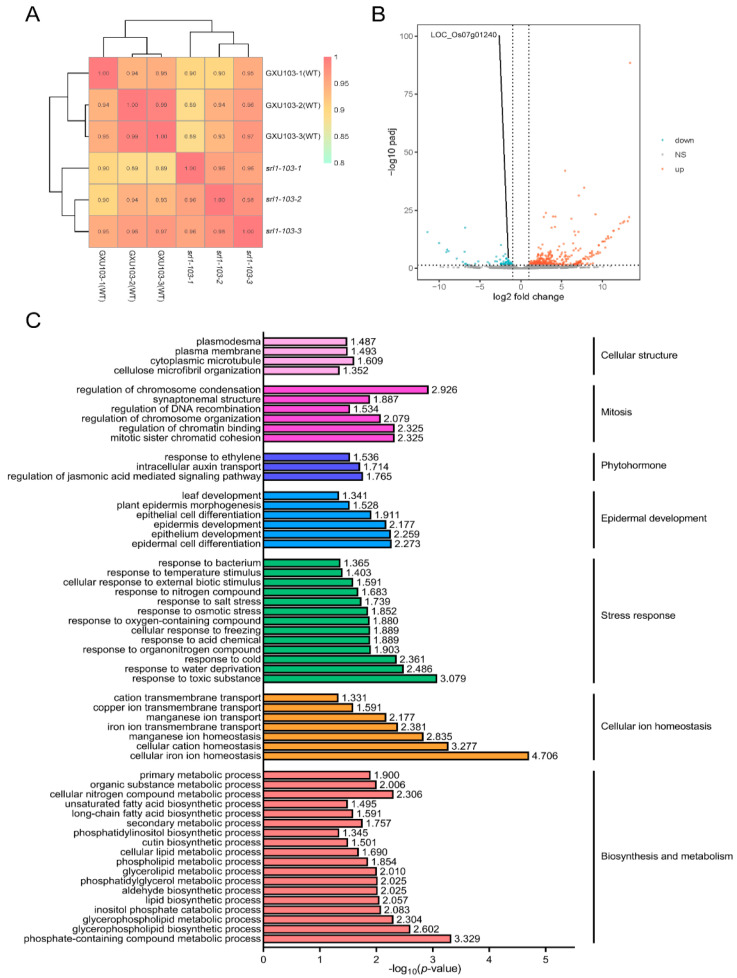
Overview of RNA-seq data analysis. (**A**) Pair-wise correlation analysis (Pearson’s correlation coefficient) of all the samples. (**B**) Genes that showed significant differential expression between GXU103 and *srl1-103*. Red dots indicate upregulated genes. Blue dots indicate downregulated genes. Gray dots indicate genes with no significant differences. (**C**) Gene ontology enrichment analysis of GXU103 and *srl1-103*.

**Figure 5 ijms-24-11087-f005:**
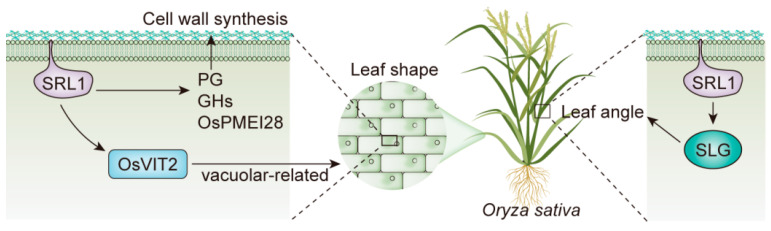
Predicted gene regulatory networks involving *SRL1*. PG: polygalacturonase. GH: glycosyl hydrolase.

**Table 1 ijms-24-11087-t001:** Analysis of mutation types in T_0_ mutant plants.

Target Site	Mutation Type									
	Homozygous		Heterozygous		Bi-Allelic		Wild		Total	
	No. of Plants	Rate (%)	No. of Plants	Rate (%)	No. of Plants	Rate (%)	No. of Plants	Rate (%)	No. of Plants	Rate (%)
Target 1	5	35.7	7	50.0	3	21.4	2	14.3	14	100.0
Target 2	4	28.6	5	35.7			5	35.7		

**Table 2 ijms-24-11087-t002:** Statistical analysis of transcriptomic sequencing data.

Sample	Total Pairs	Unique Pairs	Multiple Pairs	Alignment Rate	Unique Rate
*srl1-103-1*	22,351,256	18,287,456	1,972,979	90.65%	81.82%
*srl1-103-2*	18,852,288	15,637,550	1,014,209	88.33%	82.95%
*srl1-103-3*	20,783,805	17,707,402	867,485	89.37%	85.20%
GXU103-1 (WT)	18,877,556	15,441,394	1,594,882	90.25%	81.80%
GXU103-2 (WT)	19,015,759	16,088,667	828,635	88.96%	84.61%
GXU103-3 (WT)	18,918,633	16,199,426	670,944	89.17%	85.63%

**Table 3 ijms-24-11087-t003:** The expression changes of genes involved in leaf rolling in *srl1-103*.

MSU	MH63RS3	Type	Description	log2 Fold Change	*p* Value
*LOC_Os08g01670*	*OsMH_08G0006800*	up	invertase/pectin methylesterase inhibitor family protein	7.020593905	1.50 × 10^−5^
*LOC_Os05g46510*	*OsMH_05G0419400*	up	polygalacturonase	7.571164623	0.000102196
*LOC_Os04g51450*	*OsMH_04G0481000*	up	glycosyl hydrolases family 16	7.021122586	6.23 × 10^−5^
*LOC_Os10g28080*	*OsMH_10G0246200*	up	glycosyl hydrolase	2.793190008	0.000222432
*LOC_Os04g39814*	*OsMH_04G0369900*	up	Os4bglu9-beta-glucosidase homologue	2.279553945	1.60 × 10^−5^
*LOC_Os01g71474*	*OsMH_01G0679800*	up	glycosyl hydrolases family 17	3.361982531	3.52 × 10^−7^
*LOC_Os01g71350*	*OsMH_01G0679900*	up	glycosyl hydrolases family 17	3.325632392	0.00014919
*LOC_Os11g47600*	*OsMH_11G0449500*	up	glycosyl hydrolase	5.460140431	0.000499833
*LOC_Os09g23300*	*OsMH_09G0236200*	up	integral membrane protein	2.062419747	3.49 × 10^−6^
*LOC_Os08g44840*	*OsMH_08G0448400*	down	transferase family protein	−1.192836988	0.000195723

## Data Availability

The raw sequence data reported in this paper have been deposited in the Genome Sequence Archive [60] in the National Genomics Data Center [61] and China National Center for Bioinformation/Beijing Institute of Genomics, Chinese Academy of Sciences (GSA: CRA010568), which are publicly accessible at https://ngdc.cncb.ac.cn/gsa (accessed on 11 April 2023).

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
