# Peer review of "Improving Rice Leaf Shape Using CRISPR/Cas9-Mediated Genome Editing of *SRL1* and Characterizing Its Regulatory Network Involved in Leaf Rolling through Transcriptome Analysis"

_ijms, 2023, doi:10.3390/ijms241311087_

Round 1
Reviewer 1 Report (Previous Reviewer 3)
The authors carried out the suggested corrections, I recommend the manuscript for publication.
Language needs to be polished
Author Response
1. Language needs to be polished
Response: We apologize for the poor language in our manuscript and sincerely appreciate your suggestions on its language. In order to enhance the readability of our manuscript, we are committed to improving the language and involving native English speakers in language corrections. We genuinely hope for a substantial improvement in the language proficiency of this manuscript.

Reviewer 2 Report (Previous Reviewer 1)
The manuscript by Han et al. offers an innovative proof of concept of the use of CRISPR/Cas9 on rice for leaf shape by editing SRL1 gene. The work reads well, and definitively highlights key findings in the field of gene editing for leaf shape in rice. Figures adequately summarize major pathways and pipelines. However, before being able to recommend acceptance, I request authors to address the following amendments.
First, please explicitly define research questions and hypotheses at the end of the introduction section (L115). This will improve readability and allow readers focusing on target questions. Please also enlist in L104 the research gaps that need to be bridge as part of the research questions.
Second, authors present knockout (e.g., L254) for the gene editing strategy. Are authors using the ‘knockout’ term in replacement of the more traditional ‘off-target’ (e.g., L366) terminology within the gene editing literature? If so, please unify terminology given preference to the ‘off-target’ term, which is more standard. If not, please also report ‘off-target’ scores.
Third, it is not supporting that there are few reports of gene editing for abiotic stress, in which SRL1 actively participates (L343). Tolerance to abiotic stress is typically regarded as polygenic adaptation (many adaptive loci with subtle effect, please refer to Nat Rev Genet 2020 21:769-81, Front Plant Sci 2020 11:583323, Front Plant Sci 2018 9:128, Front Plant Sci 2018 9:1816, and PLoS One 2018 13(3):e0189597), limiting the application of gene editing. Authors should acknowledge this limitation from the very beginning of the Introduction and comment on why aim bridging these gaps. These are major topics to be revisited in the Discussion section, since they become relevant not only for teams working with rice, but also with any other orphan crops.
Fourth, and expanding L343, authors must close the Discussion section (L380) by explicitly commenting on (1) pleiotropic effects of the leaf same pathways (and the SRL1 edited gene) with other abiotic stress responses of major importance (refer to Front Genet 2020 11:564515, and Front Genet 2019 10:954), and (2) potential caveats of this study.
Last but not least, authors should close (L381) with explicit Conclusion and Perspectives sections. In the former authors should bridge the major gaps of the field (see my first comment on L380) in the light of the key achievements of the work. In the latter, authors must propose innovative ways to handle the caveats of the study (see my fourth comment, on L343, second point), and suggest key experiments and analyses that will help maximizing the potential of gene editing in rice, targeting other adaptive traits beyond leaf shape (see my fourth comment, first point). A central question that the Perspectives section should focus on is whether gene editing is feasible for highly polygenic adaptation (refer to my third comment on L343, and check Genes 2021 12:783).
Author Response
(1) First, please explicitly define research questions and hypotheses at the end of the introduction section (L115). This will improve readability and allow readers focusing on target questions. Please also enlist in L104 the research gaps that need to be bridge as part of the research questions.
Response: Thank you for providing constructive feedback on my manuscript. We have carefully considered your suggestions and made every effort to revise the manuscript accordingly. Please refer to the highlighted yellow section in the last paragraph of the "1. Introduction" for the revised content. The modifications regarding the research questions and hypotheses have been highlighted in red font, while the limitations of previous study have been highlighted in blue font. Additionally, in order to make the content of the article more coherent, we have exchanged the positions of the third and fourth paragraphs in the “1. Introduction” and made minor modifications to the content.
(2) Second, authors present knockout (e.g., L254) for the gene editing strategy. Are authors using the ‘knockout’ term in replacement of the more traditional ‘off-target’ (e.g., L366) terminology within the gene editing literature? If so, please unify terminology given preference to the ‘off-target’ term, which is more standard. If not, please also report ‘off-target’ scores.
Response: Thank you very much for your suggestions regarding the use of professional terminology in the article. I apologize for not clearly expressing the terminology, which has caused confusion for you. Regarding the word "off-target", we want to express the situation where it deviates from the target and hits other locations. We used an online tool, http://skl.scau.edu.cn/, to predict the off-target scores for the two target sites mentioned in the manuscript. The maximum off-target score for target 1 in indica rice was predicted to be 0.23, while for target 2, the maximum off-target score was predicted to be 0.144. We have supplemented this information in the highlighted yellow section at the end of the fourth paragraph in the “3. Discussion”.
(3) Third, it is not supporting that there are few reports of gene editing for abiotic stress, in which SRL1 actively participates (L343). Tolerance to abiotic stress is typically regarded as polygenic adaptation (many adaptive loci with subtle effect, please refer to Nat Rev Genet 2020 21:769-81, Front Plant Sci 2020 11:583323, Front Plant Sci 2018 9:128, Front Plant Sci 2018 9:1816, and PLoS One 2018 13(3):e0189597), limiting the application of gene editing. Authors should acknowledge this limitation from the very beginning of the Introduction and comment on why aim bridging these gaps. These are major topics to be revisited in the Discussion section, since they become relevant not only for teams working with rice, but also with any other orphan crops.
Response: Thank you very much for your advice on the involvement of SRL1 in stress tolerance. Numerous studies have consistently indicated that the regulation of stress tolerance pathways involves the intricate interplay of various genes, rendering research in this field exceptionally complex. It is difficult for an individual gene mutation to play a key role in stress tolerance, let alone the gene SRL1, which regulates leaf rolling. The focus of this article is on improving the understanding of the pathways through which SRL1 regulates leaf shape changes. Therefore, we did not discuss the aspects of stress tolerance in the introduction.
In the discussion, we analyzed the potential impact of SRL1 on resistance genes, but the specific regulatory relationships would require extensive experimentation to confirm. We are simply providing a new direction for thinking, and if this direction is correct, it would be highly relevant to the study of crop resistance. Additionally, inspired by your suggestion, we have made some modifications at the beginning of the fourth paragraph in the discussion section and highlighted these parts in yellow.
(4) Fourth, and expanding L343, authors must close the Discussion section (L380) by explicitly commenting on (1) pleiotropic effects of the leaf same pathways (and the SRL1 edited gene) with other abiotic stress responses of major importance (refer to Front Genet 2020 11:564515, and Front Genet 2019 10:954), and (2) potential caveats of this study.
Response: Thank you for providing constructive feedback on my manuscript. We have carefully considered your suggestions and made every effort to revise the manuscript accordingly. Please refer to the highlighted section in yellow in the last paragraph of the "3. Discussion" for the revised content.
(5) Last but not least, authors should close (L381) with explicit Conclusion and Perspectives sections. In the former authors should bridge the major gaps of the field (see my first comment on L380) in the light of the key achievements of the work. In the latter, authors must propose innovative ways to handle the caveats of the study (see my fourth comment, on L343, second point), and suggest key experiments and analyses that will help maximizing the potential of gene editing in rice, targeting other adaptive traits beyond leaf shape (see my fourth comment, first point). A central question that the Perspectives section should focus on is whether gene editing is feasible for highly polygenic adaptation (refer to my third comment on L343, and check Genes 2021 12:783).
Response: Thank you for providing constructive feedback on my manuscript. We have carefully considered your suggestions and made every effort to revise the manuscript accordingly. Please refer to the highlighted section in yellow in the last paragraph of the "3. Discussion" for the revised content. In the same position as the modified version of suggestion 4.

Round 2
Reviewer 2 Report (Previous Reviewer 1)
Very well defended and revised version and rebuttal letter; the work is suitable for IJMS. My last comments are to include, whenever possible, and uncertain measure in the bars of figure 4 (for instance standard error or confidence interval bars), and if figure 5 is inspired in another already published artwork, please cited within the legend.
This manuscript is a resubmission of an earlier submission. The following is a list of the peer review reports and author responses from that submission.
Round 1
Reviewer 1 Report
The report by Han et al. advances our understating of how gene editing may improve leaf shape in rice. The report is well written and condensed, as well as technically appropriate. However, before being able to recommend acceptance, I invite authors to address the following amendments.
First, reporting literature compilation at the introduction on gene editing is missing (it should go as a new paragraph after L73). For that, I encourage authors to follow PRISMA’s guidelines to compile literature in a more systematic manner (http://www.prisma-statement.org/PRISMAStatement/FlowDiagram), the above considering the boom in gene editing literature during the last year, specially in model crop plants such as rice. Please briefly comment on the concrete steps/parameters used during the search (i.e. keywords, search equation, target repositories), filtering, and summary of key references. This section can be brief (one paragraph), but yet would make the literature search for gene editing targeting more repeatable.
Second, the work is also missing a key mention that broadly summarizes how gene editing can target complex polygenic traits such as leaf shape, which is known to exhibit strong environmental effects (refer to PLoS One 2013 8(5):e62898), and a polygenic basis with many underlying loci and genes with subtle effects each (refer to Front Plant Sci 2018 9:128). This expected trend contradicts at first glance basic requirements for gene editing, including a simple genetic basis - i.e., few known genes with major effects. Therefore, authors should embrace on this bottleneck from the very begging by reporting explicit heritability scores for leaf shape, as well as overall numbers of leaf shape associated markers/genes beyond SRL1 (and their corresponding effects, refer to BMC Genet 2012 12:48). The begging of the second paragraph at L43 would be a good place for that.
Third, the introduction section should properly close (L84) with explicit research questions and hypotheses, and not just the study goal. I would recommend adding a short sentence before to emphasize the research gap that inspired pursuing this work. This will allow readers focusing on concrete research questions and expected trends on how to effectively utilize gene editing for complex traits such as leaf shape.
Fourth, since leaf shape traits are usually pleiotropic (see Agronomy 2021 11:1978), please also comment (L258 and L274 would be a good place for that since authors discuss in these sections abiotic correlates) on potential gene editing co-variation with other specific traits known to depend on leaf shape, such as heat and drought tolerance (via leaf area, stomata density and evotranspiration, refer to Front Genet 2019 10:954, Front Genet 2020 11:564515, and Genes 2021 12:556). Also related to this point on pleiotropy (and potential gene editing off-targets), authors must also briefly mention whether adaptive trade-offs for biotic resistance to leaf pathogens are observed/predicted in combination with altered leaf morphology (refer to Plants 2021 10:2022). For instance, a biomass/resistance trade-off may be evident in more subtle ways such as leaf shape and repellent compounds concentration (see Front Genet 2020 11:656). All these pleiotropic phenotypes may be potential gene editing off-targets when targeting leaf shape, something authors should acknowledge explicitly at the discussion (as a follow up to L256, where authors acknowledge covariates with resistance genes).
Last but not least, the main closing discussion paragraph (L284) should be expanded to incorporate perspectives on how to effectively merge gene editing with other modern strategies to better utilize standing variation (following Front Plant Sci 2018 9:1816), as reported here for leaf shape, within parental selection for pre-breeding schemes (refer to Theor Appl Genet 2013 126(2):535-48). The main figures in Genes 2021 12:783 and Genes 2022 13:1 may serve as guidance to better incorporate these perspectives, please refer to them.
Author Response
Response to Reviewer 1 Comments
Point 1: First, reporting literature compilation at the introduction on gene editing is missing (it should go as a new paragraph after L73). For that, I encourage authors to follow PRISMA’s guidelines to compile literature in a more systematic manner (http://www.prisma-statement.org/PRISMAStatement/FlowDiagram), the above considering the boom in gene editing literature during the last year, specially in model crop plants such as rice. Please briefly comment on the concrete steps/parameters used during the search (i.e. keywords, search equation, target repositories), filtering, and summary of key references. This section can be brief (one paragraph), but yet would make the literature search for gene editing targeting more repeatable.
Response 1: According to the reviewer's opinion, we really lack literature introduction on gene editing technology. We have added literature review on gene editing in the paper and inserted relevant references. Please see the attachment.Please see the attachment.
Point 2: Second, the work is also missing a key mention that broadly summarizes how gene editing can target complex polygenic traits such as leaf shape, which is known to exhibit strong environmental effects (refer to PLoS One 2013 8(5):e62898), and a polygenic basis with many underlying loci and genes with subtle effects each (refer to Front Plant Sci 2018 9:128). This expected trend contradicts at first glance basic requirements for gene editing, including a simple genetic basis - i.e., few known genes with major effects. Therefore, authors should embrace on this bottleneck from the very begging by reporting explicit heritability scores for leaf shape, as well as overall numbers of leaf shape associated markers/genes beyond SRL1 (and their corresponding effects, refer to BMC Genet 2012 12:48). The begging of the second paragraph at L43 would be a good place for that.
Response 2: According to the reviewer's suggestion, we have added the introduction of leaf curling gene at the beginning of the second paragraph. In the introduction of gene editing tools below, the editing of complex traits with multiple genes is also briefly summarized. Please see the attachment.
Point 3: Third, the introduction section should properly close (L84) with explicit research questions and hypotheses, and not just the study goal. I would recommend adding a short sentence before to emphasize the research gap that inspired pursuing this work. This will allow readers focusing on concrete research questions and expected trends on how to effectively utilize gene editing for complex traits such as leaf shape.
Response 3: According to the reviewer's suggestion, the introduction should clarify the research question and hypothesis at the end, rather than just the research objective. We have emphasized the importance of this research in accordance with the relevant content previously suggested. Please see the attachment.
Point 4: Fourth, since leaf shape traits are usually pleiotropic (see Agronomy 2021 11:1978), please also comment (L258 and L274 would be a good place for that since authors discuss in these sections abiotic correlates) on potential gene editing co-variation with other specific traits known to depend on leaf shape, such as heat and drought tolerance (via leaf area, stomata density and evotranspiration, refer to Front Genet 2019 10:954, Front Genet 2020 11:564515, and Genes 2021 12:556). Also related to this point on pleiotropy (and potential gene editing off-targets), authors must also briefly mention whether adaptive trade-offs for biotic resistance to leaf pathogens are observed/predicted in combination with altered leaf morphology (refer to Plants 2021 10:2022). For instance, a biomass/resistance trade-off may be evident in more subtle ways such as leaf shape and repellent compounds concentration (see Front Genet 2020 11:656). All these pleiotropic phenotypes may be potential gene editing off-targets when targeting leaf shape, something authors should acknowledge explicitly at the discussion (as a follow up to L256, where authors acknowledge covariates with resistance genes).
Response 4: In this paper, we analyzed the changes of resistance genes caused by SRL1 knockout, involving disease resistance, cold tolerance, salt tolerance, etc. The function of these resistance genes is realized by regulating cell wall synthesis, transcription factors and plant hormones. Since correlation analysis was only carried out on transcriptome data in this paper, observation and statistics were not carried out on the determination indicators of these genes. However, these indicators are essential for further studies. Secondly, it is a good idea to use the correlation between leaf morphology change and resistance or biomass proposed by the reviewer to make efficient judgment, but it requires the support of a large amount of experimental data, which is also our next important work. Finally, we have mentioned the problem of off-target gene editing in the paper.
Point 5: Last but not least, the main closing discussion paragraph (L284) should be expanded to incorporate perspectives on how to effectively merge gene editing with other modern strategies to better utilize standing variation (following Front Plant Sci 2018 9:1816), as reported here for leaf shape, within parental selection for pre-breeding schemes (refer to Theor Appl Genet 2013 126(2):535-48). The main figures in Genes 2021 12:783 and Genes 2022 13:1 may serve as guidance to better incorporate these perspectives, please refer to them.
Response 5: The combination of gene editing and other modern strategies in agricultural development can give full play to each other's advantages, improve editing efficiency and accuracy, and bring more possibilities and opportunities to agricultural production and biotechnology research. Please see the attachment. Please see the attachment.
Finally, thank you very much for the reviewer's valuable suggestions.

Reviewer 2 Report
The reason for this decision is:
This manuscript does not fulfill the standards established for the journal to be considered for publication.
First of all, I judge that this thesis is neither scientific nor factual. Because the Cas9-F/Cas9-R primers were used to regenerate the plant through the plant regeneration process. But I can't believe how there is no off-time here. Also, it is not enough to explain why transcriptome analysis is performed. I wish it was written on a more factual basis. Therefore, the manuscript contains fundamental errors that cannot be corrected through author corrections.
Author Response
Response to Reviewer 2 Comments
Point 1: First of all, I judge that this thesis is neither scientific nor factual. Because the Cas9-F/Cas9-R primers were used to regenerate the plant through the plant regeneration process. But I can't believe how there is no off-time here. Also, it is not enough to explain why transcriptome analysis is performed. I wish it was written on a more factual basis. Therefore, the manuscript contains fundamental errors that cannot be corrected through author corrections.
Response 1: Many thanks to the reviewers for their questions and valuable comments on our articles. Based on our understanding of the reviewer's comments, the following answers are given:
First, the function of primer Cas9-F/Cas9-R is to detect Cas9 protein in T1 generation plants, as shown in Figure 2C, which is to screen T-DNA-Free in T1 generation plants.
Secondly, the serial numbers 1-11 in the Figure are only the serial numbers of agarose electrophoresis samples. The results shown in Figure 2C are a small part of the results of Cas9 protein removal in the total 120 strains of the T1 generation, and are not a single strain responding to the offspring of the corresponding strains 1-11 in Figure 2B. In the T0 generation, Positive mutant plants were selected using specific primers hyg-F/hyg-R(Table S1) containing a hygromycin resistance marker, and a total of 14 positive plants were achieved (Figure 2B). The agarose gel diagram results are only part of it. n addition, the target sequence of primer SRL-F/SRL-R was amplified and sequenced to determine the gene mutation (Table S1). And the sequencing comparison results are shown in Figure 2A. All experiments were conducted on the basis of positive plant identification, target sequence analysis and Cas9 protein detection. Two stable mutant strains GXU103-1 and GXU103-5 were selected as experimental materials.Perhaps because the use of primer SRL-F/SRL-R was not mentioned in the text, and the description of the experimental results such as the screening of positive plants in T0 generation and the detection of Cas9 protein in T1 generation plants was not brief and clear, the reviewers questioned the experimental content.
In addition, as for transcriptome analysis, transcriptome sequencing is carried out to understand the interpretation of the transcriptome on the functional components of the genome, reveal the molecular components of cells and tissues, and understand development, which has been widely used in basic research and other fields. In order to further understand the role of SRL1 in the regulatory network of rice leaf development, we performed transcriptome analysis of homozygous double-target mutant srl1-103.
Finally, thank you very much for reviewers to give our precious opinion articles.

Reviewer 3 Report
Authors define the SRL1 regulatory network during leaf development, transcriptome analysis was combined with CRISPR/Cas9 to produce two targets knockout mutant of the leaf roll gene SRL1 in indica background rice.
1. Abbreviations and scientific names must be written in full in the first mention.
2. It is important to proofread this manuscript by a native English speaker
3. The abstract should contain some quantitative information.
4. Statistical analysis; Write details such as replications and software used for statistical analysis
It is important to proofread this manuscript by a native English speaker
Author Response
Response to Reviewer 3 Comments
Point 1: Abbreviations and scientific names must be written in full in the first mention.
Response 1: We have fixed the issue of full scientific names when abbreviations first appear in articles. Please see the attachment.
Point 2: It is important to proofread this manuscript by a native English speaker.
Response 2: We have modified the language expression of the article. Please see the attachment.
Point 3: The abstract should contain some quantitative information.
Response 3: As for the reviewer's suggestions on the abstract, we have carefully revised the abstract. Please see the attachment.
Point 4: Statistical analysis; Write details such as replications and software used for statistical analysis.
Response 4: According to suggestions of reviewers, we have added the part of "4.4. Statistical analysis of agronomic traits" in the article. Please see the attachment.
Finally, I would like to thank the reviewers for their valuable suggestions.

Round 2
Reviewer 1 Report
The revised paper broadly discusses the significance of rolled leaf trait in rice breeding and its impact on photosynthetic efficiency. The study focuses on the SEMI-ROLLED LEAF1 (SRL1) gene, which encodes a glycosylphosphatidylinositol-anchored protein involved in regulating leaf rolling. The researchers employed modern CRISPR/Cas9 technology to enhance the leaf rolling trait in the indica rice line GXU103.
The authors successfully edited the second and third exons of the SRL1 gene using CRISPR/Cas9, resulting in the generation of a T-DNA-free double targets mutant line called srl1-103 in the T1 generation. Compared to the wild type, the srl1-103 line exhibited distinct characteristics, including developmental defects, rolled upright leaves, and a more compact plant morphology.
To gain insights into the role of SRL1 in the regulatory network of rice leaf development, the researchers further performed transcriptome analysis on homozygous double target mutant strains within the srl1-103 line. The analysis revealed 459 differentially expressed genes between the mutant and wild type plants, with 388 upregulated genes and 71 downregulated genes. These differentially expressed genes were found to be significantly enriched in cell wall synthesis and vacuolar-related processes. The altered expression of these genes may contribute to the leaf rolling phenotype observed in the srl1-103 mutant.
Furthermore, the study now highlights more clearly the significant downregulation of the BAHD acyltransferase-like protein gene as a potential driverfor the reduced leaf inclination and compact nature of the srl1-103 mutant.
In summary, this revised paper provides valuable insights into the functional role of the SRL1 gene in rice leaf rolling and demonstrates the successful use of CRISPR/Cas9 technology to improve this agronomic trait. The transcriptome analysis elucidates a regulatory network involving SRL1 and identifies key genes associated with cell wall synthesis and vacuolar functions. These findings contribute to the understanding of rice leaf development and provide a theoretical basis for utilizing the SRL1 gene in rice pre- and breeding programs aimed at improving leaf rolling traits and variety enhancement.
Overall, the paper, after the revisions made by the authors in the light of the reviewers' comments, represents a well-structured study with clear objectives, and the findings provide important contributions to the field of rice genetics and breeding.
However, authors must still (1) build a more comprehensive discussion of the functional validation and phenotypic characterization of the mutant line, as well as (2) prospect within the perspectives more concrete implications for crop improvement. This amendments, encourage from the previous revision, would further enhance the impact of the research.
Author Response
Response to Reviewer 1 Comments
Point 1: Build a more comprehensive discussion of the functional validation and phenotypic characterization of the mutant line
Response 1: Thank you very much for the reviewer's suggestions. We have made the necessary revisions to the manuscript based on the reviewer's requirements. Please see the attachment.
Point 2: Prospect within the perspectives more concrete implications for crop improvement.
Response 2: Thank you very much for the reviewer's suggestions. We have made the necessary revisions to the manuscript based on the reviewer's requirements. Please see the attachment.

Reviewer 2 Report
Please note that this journal cannot be printed because the number of plant regeneration populations and the agricultural characteristics of the corresponding plants are not compared and described. In addition, many studies have been conducted on the angle of the leaves at rice, so please do a lot of discussion.
Author Response
Response to Reviewer 2 Comments
Point 1: Please note that this journal cannot be printed because the number of plant regeneration populations and the agricultural characteristics of the corresponding plants are not compared and described.
Response 1: Based on your inquiry, we have carefully reviewed the entire article. We have identified a foolish mistake in section “4.4. Statistical analysis of agronomic traits”, where the experimental material was incorrectly referred to as T1 generation. This error occurred during the hurried revision process to address suggestions from other reviewers, and we apologize for the oversight in not thoroughly proofreading. The experimental materials used for statistical analysis of agronomic traits, phenotypic analysis, and transcriptome analysis in this study are stable populations of the T2 generation. The reviewer has raised concerns regarding the content and data of the manuscript due to this oversight. The relevant information has been revised and supplemented in the main text and the corresponding section of the experimental methods.We greatly appreciate the reviewer for promptly identifying this problem, as it has provided us with an opportunity to rectify the error.We sincerely appreciate the reviewer for bringing this matter to our attention and giving us the opportunity to rectify it.
Point 2: In addition, many studies have been conducted on the angle of the leaves at rice, so please do a lot of discussion.
Response 2: Thank you very much for the reviewer's suggestions. According to the suggestions of the reviewer, we have modified the article.Please see the attachment.

Round 3
Reviewer 2 Report
I have reviewed the paper, but once again notice that it cannot be reprinted. The revised paper confirms that it has not changed significantly.